# Bloch-Lorentz magnetoresistance oscillations in delafossites

Kostas Vilkelis[1,2*], Lin Wang[2†] and Anton Akhmerov[2]

**1** Qutech, Delft University of Technology, Delft 2600 GA, The Netherlands
**2** Kavli Institute of Nanoscience, Delft University of Technology,
Delft 2600 GA, The Netherlands

⋆ kostasvilkelis@gmail.com

## Abstract

Recent measurements of the out-of-plane magnetoresistance of delafossites ($PdCoO_2$ and $PtCoO_2$) observed oscillations closely resembling the Aharonov-Bohm effect. Here, we show that the magnetoresistance oscillations are explained by the Bloch-like oscillations of the out-of-plane electron trajectories. We develop a semiclassical theory of these Bloch-Lorentz oscillations and show that they are a consequence of the ballistic motion and quasi-2D dispersion of delafossites. Our model identifies the sample wall scattering to be the most likely factor limiting the visibility of these Bloch-Lorentz oscillations in existing experiments.



## Contents

---

† Current adress: Department of Physics, University of Konstanz, D-78457 Konstanz, Germany

# 1 Introduction

Known since the discovery of mineral $CuFeO_2$ by Friedel in 1873, delafossites are materials with the general formula $ABO_2$ [1,2]. Delafossites are naturally occurring layered structures of alternating conductive A layer and insulating $BO_2$ layer with the overall $R\bar{3}m$ space group [3]. These materials are considered to be 2D owing to their weak interlayer coupling which results in a nearly cylindrical Fermi surface [4,5]. Of particular interest are $PdCoO_2$ and $PtCoO_2$ which were first synthesized and characterized at room temperature in 1971 by Shannon *et al.* [2,3, 6]. Even though nearly 50 years have passed since then, the area of research is still very active due to the delafossites' impressive electronic transport properties [7]. At room temperature, it was shown that the conductivity of $PdCoO_2$ is $2.6\,\mu\Omega\,cm$, very close to that of elemental copper [8]. Part of the reason for such large conductivity is the high Fermi velocity $7.5\times10^5\,m\,s^{-1}$ [8]. Another reason is their exceptional mean-free path at $4\,K$ which exceed $20\,\mu m$ [8]. Such value of mean-free path is accredited mostly to anomalously clean nature of delafossites and orbital-momentum locking [9,10]. Overall, all of these properties of delafossites make them a good platform to study mesoscopic ballistic transport [11].

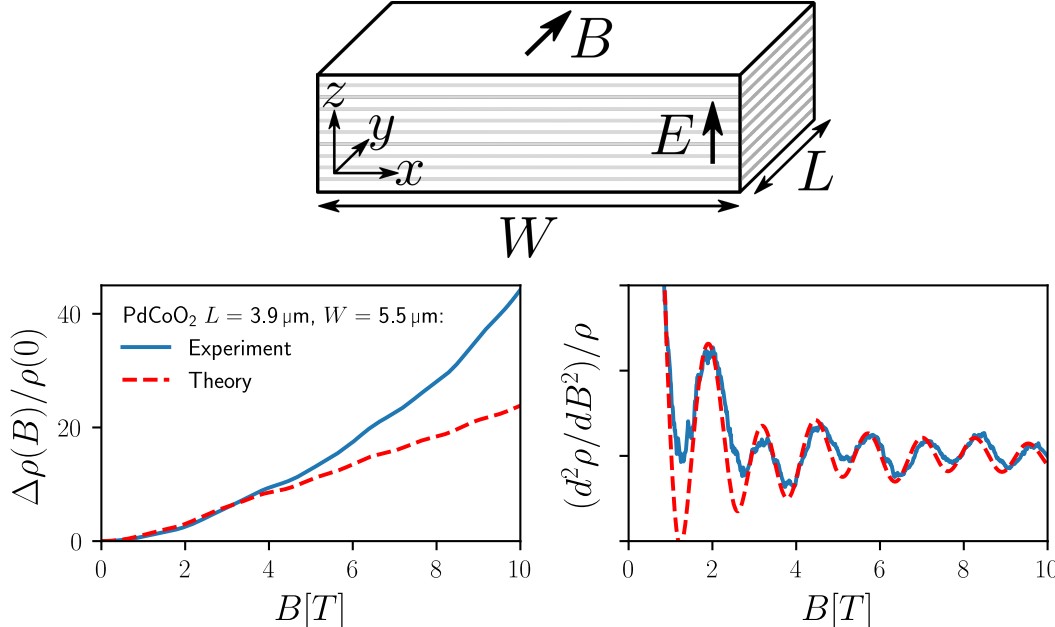

Figure 1: $PdCoO_2$ magnetoresistance experimental set up (top) and results (solid blue lines) obtained by Putzke *et al.* [12]. The semiclassical prediction (dashed red lines) was obtained by modeling the finite size $PdCoO_2$ sample.

Recent experiments studied the out-of-plane transport of $PdCoO_2$ [12], with the setup and the measured magnetoconductance shown in Fig. 1. Magnetoconductance was measured with the magnetic field applied in the plane of the delafossite layers and the current passing out-of-plane. Surprisingly, the magnetoconductance showed oscillations with a magnetic field similar to the Aharonov-Bohm effect, and therefore appearing to be of quantum origin. The period of the oscillations corresponded to adding a flux quantum through the area $Wc$ with $W\sim5\,\mu m$ the width of the sample and $c$ the spacing between the adjacent conducting layers. Given that the oscillations persist up to elevated temperature of $50\,K$, this result is remarkable compared with *e. g.* quantum Hall interferometers, where coherence at micron length scale vanishes below $100\,mK$ [13].

Simulations performed by Putzke *et al.* confirmed that the oscillations of Kubo conductivity with Aharonov-Bohm periodicity indeed appear in a minimal tight-binding model that combines high anisotropy with magnetic field [12]. Based on this observation, Ref. [12] attributed the oscillations to long-range coherence of the delafossite layers ($l_\phi \geq 10\,\mu m$) and separately ruled out multiple semiclassical or macroscopic origins of these oscillations. The manuscript therefore opens a question about the possible origins of this unusual long length and high temperature phase coherence. The coherent origins of the phenomenon also require closed trajectories, and are therefore hard to reconcile with boundary scattering at the strongly disordered sample boundaries.[1]

Here, we argue that contrary to the claim of long range coherence, the oscillations are a consequence of the shape of the semiclassical trajectories rather than an interference pattern of electron waves traversing the sample. Our construction extends the idea put forward by Pippard [14] used to explain magnetoresistance oscillations in gallium [15]. Our formalism does not rely on sample-scale phase coherence and therefore is compatible with the low temperature phase coherence length $l_\phi = 400\,nm$ estimated from Shubnikov-de Haas [12] experiments. The semiclassical approach also allows us to incorporate the appropriate bulk and boundary scattering rates, and simulate the full 2D cross-section of the sample. The semiclassical approach also allows us to isolate the role of different scattering mechanisms and to conclude that in the clean samples, the sample aspect ratio is the most likely factor limiting the visibility of the oscillations.

## 2 Ballistic in-plane model in the weak out-of-plane coupling limit

Delafossites' conduction band is well approximated by the energy dispersion

$$\varepsilon(\mathbf{k}) = \varepsilon_\parallel\left(\mathbf{k}_\parallel\right) - t_z \cos(k_z c'), \tag{1}$$

where $\varepsilon_\parallel(\mathbf{k}_\parallel)$ is the in-plane dispersion relation with an approximately hexagonal Fermi surface [8,16,17], $c'$ is the interlayer distance and $t_z$ is the interlayer hopping. While interlayer dispersion is weak [4,8]—$t_z \sim 10\,meV$ is much smaller than the in-plane bandwidth—it exceeds the thermal broadening of the Fermi surface at temperatures $T \lesssim 50\,K$. This motivates a perturbative approach in terms of $t_z$ that we use throughout the paper.

We compute electron density $f(\mathbf{r}, \mathbf{k}, t)d^3\mathbf{k}$ at position $\mathbf{r}$, time $t$ and momentum $\mathbf{k}$ by using the Boltzmann equation. We separate electron density into the equilibrium part and non-equilibrium parts:

$$f(\mathbf{r}, \mathbf{k}) = f^0 - g(\mathbf{r}, \mathbf{k})\frac{\partial f^0}{\partial \varepsilon}, \tag{2}$$

where the equilibrium density $f^0$ (Fermi-Dirac distribution) becomes at zero temperature a Heaviside function so that its derivative $\frac{\partial f^0}{\partial \varepsilon}$ becomes a Dirac delta function centered around the Fermi energy. The resulting steady-state linearised Boltzmann equation [18] reads:

$$\mathbf{v} \cdot \nabla_\mathbf{r} g - \frac{e}{\hbar}(\mathbf{v} \times \mathbf{B}) \cdot \nabla_\mathbf{k} g - e v_z \mathcal{E}_z = \mathcal{L}g, \tag{3}$$

where $\mathbf{v}$ is the velocity, $e$ is the elementary charge, $\hbar$ is the reduced Planck constant, $\mathbf{B}$ is the magnetic field, $\mathcal{E}_z$ is the electric field along the out-of-plane direction, and $\mathcal{L}g$ is the linearised collision integral. The boundary conditions at the boundary coordinate $\mathbf{r}_B$ are

$$|\mathbf{v}\left(\mathbf{k}_\parallel\right) \cdot \hat{\mathbf{n}}_B| g(\mathbf{r}_B, \mathbf{k}) = \int_{\mathbf{v}(\mathbf{k}_\parallel') \cdot \hat{\mathbf{n}}_B > 0} K\left(\mathbf{k}', \mathbf{k}\right) \times \left|\mathbf{v}(\mathbf{k}_\parallel) \cdot \hat{\mathbf{n}}_B\right| g(\mathbf{r}_B, \mathbf{k}')d^3k', \qquad \mathbf{v}(\mathbf{k}_\parallel) \cdot \hat{\mathbf{n}}_B < 0, \tag{4}$$

---

[1]The samples in Ref. [12] were produced using focused ion beam lithography, and therefore have a few nm amorphous layer at the boundary.

where $\hat{\boldsymbol{n}}_B$ is the unit normal vector of the boundary (pointing inwards) and $K$ is the boundary scattering kernel.

Utilizing the smallness of $t_z$, we expand $g$ in Eq. (3) as a series to first order in $t_z$:

$$g(\boldsymbol{r},\boldsymbol{k}) \approx g_0(\boldsymbol{r},\boldsymbol{k}) + g_1(\boldsymbol{r},\boldsymbol{k}), \tag{5}$$

where $g_0$ does not depend on $t_z$ and $g_1 \propto t_z$. With the magnetic field inside the $yz$-plane $\boldsymbol{B} = (0, B_y, B_z)$, the zeroth-order expansion is

$$\boldsymbol{v}_{\parallel} \cdot \frac{\partial g_0}{\partial \boldsymbol{r}_{\parallel}} - \frac{e}{\hbar}(\boldsymbol{v}_{\parallel} \times \boldsymbol{B}) \cdot \boldsymbol{\nabla}_{\boldsymbol{k}} g_0 = \mathcal{L} g_0, \tag{6}$$

where $\boldsymbol{v}_{\parallel}$ is in-plane velocity. Equation (6) describes an electron in a magnetic field with no external forces capable of generating a steady non-equilibrium distribution. Under these conditions, non-zero scattering ensures that the steady state solution is $g_0 = 0$. Therefore, to first order in $t_z$ linearised Boltzmann equation is

$$\boldsymbol{v}_{\parallel} \cdot \frac{\partial g}{\partial \boldsymbol{r}_{\parallel}} - \frac{e}{\hbar}\left(\boldsymbol{v}_{\parallel} \times \boldsymbol{B}\right) \cdot \boldsymbol{\nabla}_{\boldsymbol{k}} g - e v_z \mathcal{E}_z = \mathcal{L} g. \tag{7}$$

Additionally, since $g \propto t_z$, it is sufficient to approximate $\mathcal{L}$ to zeroth order in $t_z$.

Integrating Eq. (7) over $k_z$ within the 1st Brillouin zone, we obtain an equation identical to Eq. (6), but with $g_0$ replaced by $g_{\parallel}(\boldsymbol{r},\boldsymbol{k}_{\parallel}) \equiv \int_{\mathrm{BZ}} g(\boldsymbol{r},\boldsymbol{k}) dk_z'$. Therefore, the in-plane current of electrons vanishes in the steady state:

$$\int_{BZ} g(\boldsymbol{r},\boldsymbol{k}) dk_z = 0. \tag{8}$$

We assume that the disorder in the bulk and at the boundaries is weakly correlated across the layers. Therefore, the disorder rapidly randomizes out-of-plane momentum $k_z$ and leads to $k_z$-independent $K$ and $\mathcal{L}$. The weakly correlated disorder together with Eq. (8) simplifies the linearised collision integral:

$$\mathcal{L} g = -\frac{g(\boldsymbol{k}_{\parallel}, k_z)}{\tau(\boldsymbol{k}_{\parallel})}, \tag{9}$$

where $\tau$ is the relaxation time which depends only on the in-plane wavevector $\boldsymbol{k}_{\parallel}$. Similarly, substitution of Eq. (8) in Eq. (4) and using the independence of $K$ from $k_z$ yields the simplified boundary conditions:

$$g(\boldsymbol{r}_B, \boldsymbol{k}_{\parallel}, k_z) = 0, \quad \text{for} \quad \boldsymbol{v}(\boldsymbol{k}_{\parallel}) \cdot \hat{\boldsymbol{n}}_B < 0. \tag{10}$$

Neither the scattering Eq. (9) nor boundary conditions Eq. (10) mix non-equilibrium electron densities along different trajectories defined by the semiclassical equations of motion

$$\hbar \frac{d \boldsymbol{r}(t)}{dt} = \boldsymbol{\nabla}_{\boldsymbol{k}} \varepsilon(\boldsymbol{k}), \quad \hbar \frac{d \boldsymbol{k}(t)}{dt} = -e \boldsymbol{v}(t) \times \boldsymbol{B}, \tag{11}$$

where $t$ is the time along the trajectory. Therefore, using Eq. (11), we obtain the evolution of $g$ along a single trajectory:

$$\frac{\partial g(t, \boldsymbol{r}_0, \boldsymbol{k}_0)}{\partial t} = \boldsymbol{v} \cdot \boldsymbol{\nabla}_{\boldsymbol{r}} g - \frac{e}{\hbar}(\boldsymbol{v} \times \boldsymbol{B}) \cdot \boldsymbol{\nabla}_{\boldsymbol{k}} g. \tag{12}$$

Here we parameterize each trajectory originating at a sample boundary through its initial coordinate and wave vector $\boldsymbol{r}_0 = (x_0, y_0, z_0)$ and $\boldsymbol{k}_0 = (k_0 \cos\theta_0, k_0 \sin\theta_0, k_{z,0})$. Substituting Eq. (12) and Eq. (9) into Eq. (7), we obtain the Boltzmann equation along a single trajectory

$$\frac{\partial g(t, \boldsymbol{r}_0, \boldsymbol{k}_0)}{\partial t} - e\mathcal{E}_z v_z(k_z(t)) = -\frac{g(t, \boldsymbol{r}_0, \boldsymbol{k}_0)}{\tau(\boldsymbol{k}_{\parallel}(t))}, \tag{13}$$

with solution

$$
\begin{aligned}
g(t, \boldsymbol{r_0}, \boldsymbol{k_0}) &= -e\mathcal{E}_z \int_0^t v_z\left(k_z(t')\right) \times \exp\left(-\frac{(t-t')}{\tau(\boldsymbol{k}_\parallel(t))}\right) dt' \\
&= -\frac{e\mathcal{E}_z t_z}{\hbar} \mathrm{Re}\left[\exp\left(ik_{z,0}\right) \int_0^t \exp\left(i\Delta k_z(t)\right) \times \exp\left(-\frac{(t-t')}{\tau(\boldsymbol{k}_\parallel(t))}\right) dt'\right], \quad (14)
\end{aligned}
$$

where $\Delta k_z(t)$ is the $k_z(t)$ solution to Eq. (11) with $k_z(0) = 0$ initial condition. Because $\Delta k_z(t)$ is fully determined by the in-plane trajectories, Eq. (14) shows that the excess electron distribution $g$ is also fully determined by the in-plane trajectories.

To analyze experimental observations, we compute the current along $z$

$$
I_{zz} = e \int_{S_\parallel} d^2\boldsymbol{r}_\parallel \iiint_{\mathrm{BZ}} f(\boldsymbol{r}, \boldsymbol{k}) v_z(\boldsymbol{k}) d\boldsymbol{k}, \quad (15)
$$

where the triple integral is over the 1st Brillouin zone, and $S_\parallel$ is the in-plane surface area of the sample. We express the out-of-plane conductivity $\sigma_{zz} = I_{zz}/(S_\parallel \mathcal{E}_z)$ at zero temperature by substituting Eq. (2) into Eq. (15):

$$
\sigma_{zz} = \frac{e}{S_\parallel \mathcal{E}_z} \int_{S_\parallel} d^2\boldsymbol{r}_\parallel \iiint_{\mathrm{BZ}} \delta(\varepsilon - \varepsilon_F) g(x, \boldsymbol{k}) v_z(\boldsymbol{k}) d\boldsymbol{k}, \quad (16)
$$

where $\varepsilon_F$ is the Fermi energy. In order to compute the lowest nonvanishing contribution in $t_z$ to conductivity, we use $g_0 = 0$, and approximate the energy $\varepsilon(k_x, k_y, k_z) \approx \varepsilon_\parallel$ only to zeroth order in $t_z$. We switch to cylindrical coordinates in k-space $(x, k, \theta, k_z)$ where $k$ is the in-plane wavevector length and $\theta$ is the azimuth. The conductivity to the lowest order in $t_z$ is

$$
\sigma_{zz} = \frac{e}{S_\parallel \mathcal{E}_z \hbar} \int_{S_\parallel} \int_0^{2\pi} \int_{-\frac{\pi}{c'}}^{\frac{\pi}{c'}} \frac{k_F(\theta)}{v_R(\theta)} g(x, k_F(\theta), \theta, k_z) \times v_z(k_z) d^2\boldsymbol{r}_\parallel d\theta dk_z, \quad (17)
$$

where

$$
v_R(\theta) = \frac{1}{\hbar} \frac{\partial \varepsilon_\parallel}{\partial k}, \quad (18)
$$

and $k_F(\theta)$ is the Fermi wavevector $\varepsilon(k_F(\theta), \theta) = \varepsilon_F$. We further simplify Eq. (17) by using that $k_z$ enters Eq. (14) as a single complex exponent and carry out integration over $k_z$ in a closed form.

## 3 Results

### 3.1 Large aspect ratio limit

Because the mean free path of delafossites is larger than the sample size [12], to illustrate the origin of the oscillations we first neglect scattering $\mathcal{L}g = 0$. Furthermore, we assume the sample has a large aspect ratio $L/W \to \infty$ and therefore we utilize translational invariance of the sample along the $y$-direction. With in-plane magnetic field $\mathbf{B} = (0, B_y, 0)$, the Boltzmann Eq. (7) reduces to

$$
v_x \frac{\partial g(x, v_x, k_z)}{\partial x} - v_x \frac{eB_y}{\hbar} \frac{\partial g(x, v_x, k_z)}{\partial k_z} - e\mathcal{E}_z v_z = 0. \quad (19)
$$

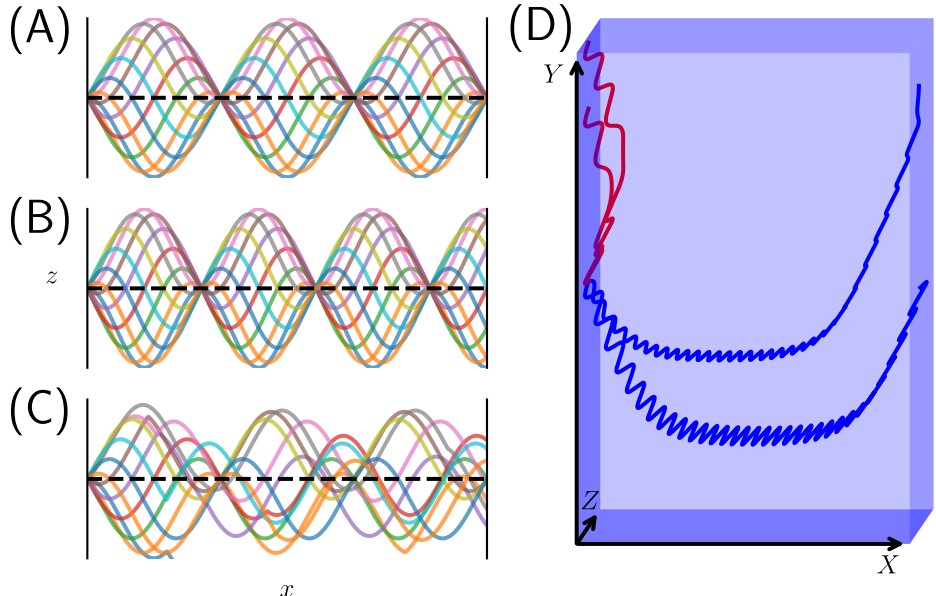

Figure 2: **(A)** Trajectories with oscillations commensurate to the sample width due to an in-plane magnetic field. Different curves indicate the different initial phases of the trajectory. **(B)** Same as in (A), but the in-plane magnetic field is chosen to give incommensurate oscillations. **(C)** Same as in (A), but with scattering present. **(D)** Trajectories due to an out-of-plane magnetic field. Blue lines are boundary-to-boundary trajectories whereas the red lines are edge-localized trajectories. Only the boundary-to-boundary trajectories produce current oscillations due to a net $k_z$ drift throughout the trajectory.

In this simple limit, $g(x, v_x, k_z)$ only depends on $k_x$ and $k_y$ through $v_x(k_x, k_y)$. Solution to Eq. (19) fulfilling the boundary conditions of Eq. (10) is

$$g(x, v_x, k_z) = \frac{-t_z \mathcal{E}_z}{B_y v_x}\left[\cos(k_z c') - \cos\left(k_z c' + \frac{\omega B_y}{W} x_B\right)\right],\tag{20}$$

with:

$$\omega = \frac{e}{\hbar} c' W, \quad x_B = \begin{cases} x, & \text{for } v_x > 0, \\ x - W, & \text{for } v_x < 0. \end{cases}\tag{21}$$

We substitute Eq. (20) into Eq. (17), and obtain the conductivity along $z$

$$\sigma_{zz} = \frac{e\pi t_z^2}{\omega \hbar B_y^2}\left(1 - \cos(\omega B_y)\right)\int_0^{2\pi} \frac{k_F(\theta)}{v_x(\theta) v_R(\theta)} d\theta.\tag{22}$$

In other words, the conductivity has oscillations with an experimentally observed periodicity, but it vanishes in the minima so that the oscillations have a much larger amplitude.

To explain the large amplitude of the oscillations, we consider electron trajectories. When the magnetic field is of the form $\mathbf{B} = (0, B_y, B_z)$, the $k_z$ dependence on x is

$$k_z(x) = k_{z0} + \frac{e}{\hbar} B_y x.\tag{23}$$

This ensures that all trajectories have a similar oscillatory vertical displacement as a function of $x$:

$$z(x) = \frac{t_z}{\hbar v_x}\left[\cos\left(k_{z0} + \frac{e}{\hbar} B_y x\right) - \cos(k_{z0})\right].\tag{24}$$

We plot the trajectories in Fig. 2(A, B). The universal trajectory shape is a result of the $k_z$ advancing over the complete out-of-plane Brillouin zone, similar to Bloch oscillations [19], however, the origin of the momentum drift is Lorentz force instead of the electric field. This gives $k_z$ a universal dependence on $x$ regardless of the in-plane trajectory. When the oscillation period is commensurate with the sample width, all trajectories have a zero net vertical displacement over the time of flight, and therefore carry no current as shown in Fig. 2(A). At the same time, the vertical displacement of different trajectories—and therefore the current— is maximal when a half-integer number of oscillation periods fits into the sample width as shown in Fig. 2(B). Because the contribution of every trajectory to the conductance has the same magnetic field dependence, as seen in Eq. (22), this minimal model yields an oscillatory conductance with a correct frequency, but full visibility of the oscillations in contrast to the experimental data. We define the overall phenomenon as Bloch-Lorentz oscillations.

### 3.2 Realistic sample geometry

Bulk scattering cannot explain the disagreement between the experiment and the minimal model because the mean-free-path of 20 μm [8] is much larger than the dimensions of samples used by Putzke *et al.* [12] (4 μm to 6 μm). Therefore, the dominant source of scattering must originate from the boundaries. In the experimental setup by Putzke *et al.* [12], the sample has a low aspect ratio with a sample length shorter than the width $W > L$. As a result, we expect the boundaries along the length of the sample to alter the semiclassical trajectories.

To analyse the effects of small aspect ratio, we consider a rectangular geometry with boundaries at: $x = 0$, $x = W$, $y = 0$, $y = L$. We parameterize the trajectories by their point of origin at the boundary and the initial angle $\theta_0$. At a sufficiently high out-of-plane magnetic field, bulk cyclotron orbits appear that do not intersect with sample boundaries. We disregard these trajectories because they do not contribute to the $h/e$ magnetoresistance oscillations, however extending our approach to those trajectories is straightforward. By changing the variables in Eq. (17) to the trajectory coordinate system $(t, \theta_0, k_{z0})$, we bring $\sigma_{zz}$ to the form

$$
\sigma_{zz} = \frac{-e}{W\mathcal{E}_z\hbar} \oint dr_0 \int_{\theta_{min}}^{\theta_{max}} d\theta_0 \int_0^{t_B(\theta_0,\mathbf{r_0})} dt' \frac{k_F(\theta(\theta_0,t'))}{v_R(\theta(\theta_0,t'))} J(t',\theta_0)
$$
$$
\times \int_{-\pi/c}^{\pi/c} dk_{z0}\, g(\mathbf{r_0},t',\theta_0,k_{z0}) v_z(\mathbf{r_0},t',\theta_0,k_{z0}), \quad (25)
$$

with the Jacobian determinant:

$$
J(t',\theta_0) = \left(\frac{\partial \theta}{\partial t'}\bigg|_{t'=0}\right)^{-1} \frac{\partial \theta}{\partial t'} v_x(0,\theta_0). \quad (26)
$$

In Eq. (25), the $r_0$ integral is over the sample boundary and $t_B(\theta)$ is the time that the trajectory hits a boundary. The integral over $\theta_0$ includes the contributions of all trajectories within the sample.

We solve Eq. (25) numerically with an in-plane magnetic field $B_y$ and without bulk scattering $\tau \to \infty$. The results in Fig. 3 left panel shows the oscillations decaying with decreasing aspect ratio $L/W$ of the sample. The visibility of the oscillations drops with a lower aspect ratio due to more trajectories scattering off the sample side-boundaries. Using the geometry of the sample of Ref. [12], we confirm that the computed relative magnitude of the oscillations agrees with the measured values, however, the overall resistance profile is somewhat

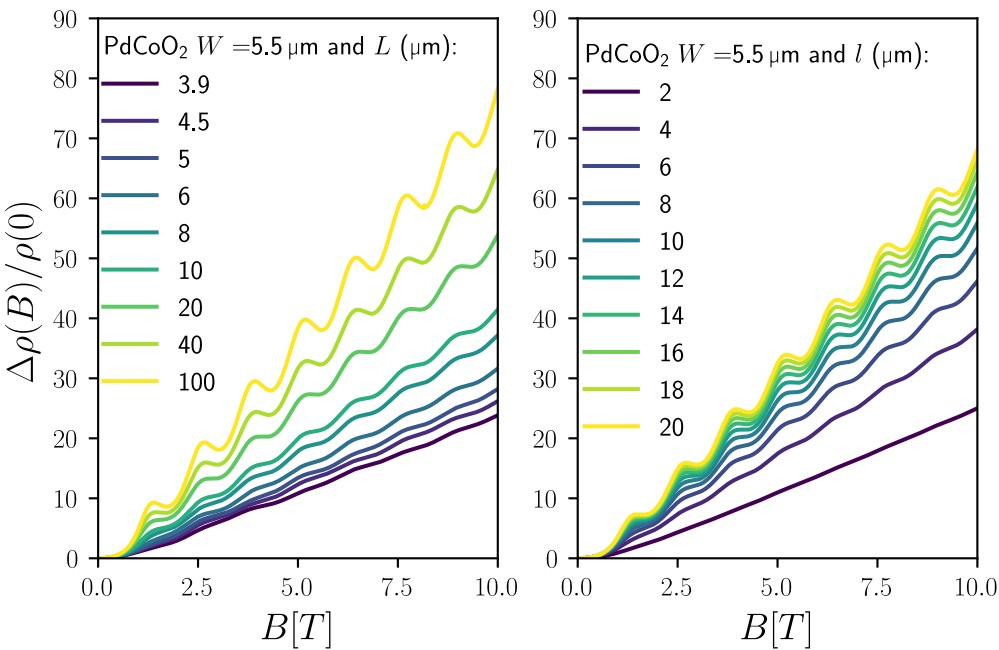

Figure 3: (Left panel) Semiclassical predictions of PdCoO$_2$ magnetoresistance with variable sample aspect ratio and no bulk scattering. (Right Panel) Semiclassical predictions of PdCoO$_2$ magnetoresistance sample with translational invariance along $y$ and variable bulk scattering (mean-free-path $l$).

different, as shown in Fig. 1. The possible reasons for this disagreement are residual bulk scattering, minor misalignment of the magnetic field, or inhomogeneity of the sample along the $z$-direction.

The scattering from the side-boundaries plays a similar role to bulk disorder. To demonstrate this, we apply the theory in the high aspect ratio limit in Eq. (19) to include bulk scattering through Eq. (9). The solution to the resulting linearised Boltzmann equation with relaxation is

$$g(x, \theta, k_z) = \frac{\mathcal{E}_z \tau e c' t_z}{\hbar \left(B_y^2 \phi^2 + 1\right)} \left( B_y \phi \cos\left(k_z c'\right) + \sin\left(k_z c'\right)\right.$$
$$\left. - \exp\left(-\frac{x_B}{l}\right)\left[ B_y \phi \cos\left(k_z c' + \frac{\omega B_y}{W} x_B\right) + \sin\left(k_z c' + \frac{\omega B_y}{W} x_\theta\right)\right]\right), \quad (27)$$

with

$$l(\theta) = \tau v_x(\theta), \quad \phi(\theta) = \frac{e}{\hbar} c' l(\theta). \quad (28)$$

Here we assume that $\tau$ is constant along the Fermi surface. We substitute Eq. (27) into Eq. (17), and obtain conductivity per unit azimuth

$$\sigma_{zz}(B_y) = \frac{\tau e^2 t_z^2 c' \pi}{\hbar^2} \int_0^{2\pi} \frac{k_F(\theta)}{v_R(\theta)\left(B_y^2 \phi^2 + 1\right)^2} \times \left(1 - r + (B_y\phi)^2 r + (B_y\phi)^2 + r \exp\left(-\frac{1}{r(\theta)}\right)\right.$$
$$\left. \times \left[(1 - B_y^2\phi^2)\cos(\omega B_y) - 2B_y\phi\sin(\omega B_y)\right]\right) d\theta, \quad r \equiv l(\theta)/W. \quad (29)$$

We recover a simple Drude model $B^2$ resistivity scaling [20] in Eq. (29) by removing the boundaries, $W \to \infty$, which removes the second term in Eq. (29). The results of Eq. (29) for various

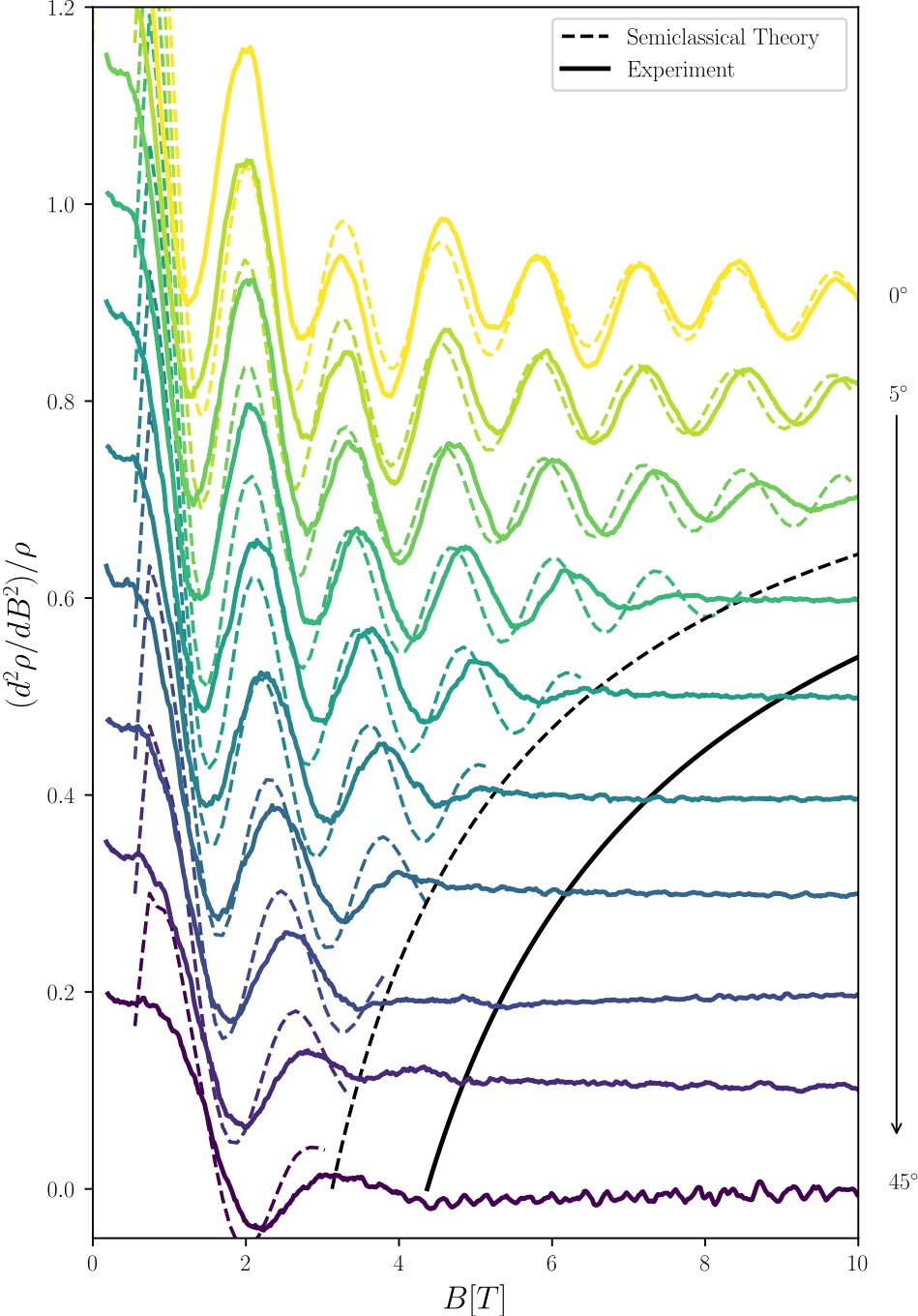

Figure 4: Numerical results from the semiclassical theory (dashed lines) with $l(0) = 4.4\,\mu m$ compared to the experimental (solid lines) magnetoresistance results by Putzke *et al.* [12] for magnetic field tilted out-of-plane by 5° steps. Black lines indicate the critical field when the cyclotron orbit fits inside the sample. The critical field value is determined by the shorter side of the sample. In the experiment, this is the length of the sample $L$, whereas in the semiclassical prediction it is the width of the sample $W$.

values of mean-free-path $l$ are shown in the right panel of Fig. 3. We observe that the scattering of the side boundaries in a sample with a finite aspect ratio results in a similar magnetoresistance as bulk scattering. Moreover, our simulations show that the magnitude of the oscillations due to side boundary scattering in the geometry used in the experiment is comparable to the observed one (see Fig. 1). Based on this we conclude that the sample aspect ratio is the factor likely limiting the oscillation visibility in the experiment.

### 3.3 Out-of-plane magnetic field

In the presence of an out-of-plane magnetic field $B_z$, the in-plane projection of each trajectory is a rotated and rescaled hexagonal Fermi surface, while the out-of-plane motion follows the oscillatory pattern of Eq. (24) (see Fig. 2(D)). We use the integral form of the Boltzmann Eq. (25) to find the magnetoresistance response. To reduce the numerical cost required to evaluate a 4D integral of Eq. (25), we approximate the side boundary scattering by using a finite relaxation time $\tau$ instead. We expect that this approximation, while somewhat crude, should capture the essential physics, as supported by the comparison between the two mechanisms shown in Fig. 3. To evaluate the remaining 3D integral, we choose the starting point of each trajectory as $t = 0$, so that its initial conditions are $\boldsymbol{r_0} = (x_0, y_0, z_0)$ and $\boldsymbol{k_0} = (k_0 \cos \theta_0, k_0 \sin \theta_0, k_{z,0})$. Here $x_0 = 0$ and $-\pi/2 < \theta_0 < \pi/2$ at the left boundary, while $x_0 = W$ and $\pi/2 > \theta_0 > 3\pi/2$ at the right boundary.

In presence of the out-of-plane magnetic field, some trajectories cross from one boundary to the opposite, while others return to the boundary from which they originated, as shown in Fig. 2(D). Only the trajectories that cross the sample contribute to the conductance oscillations because they have a net $k_z$ drift given by Eq. (23). On the other hand, the trajectories returning to the boundary where they originated do not contribute to the oscillations. As Putzke *et al.* [12] pointed out, once the cyclotron orbits become smaller than $W$, which happens at

$$B_z > \left( \frac{2\hbar k_F}{eW} \right), \tag{30}$$

with $k_F$ the Fermi wavevector, ballistic trajectories crossing the sample disappear, and so do the conductance oscillations.

We perform numerical integration of Eq.(25), with the result shown in Fig. 4. The model qualitatively agrees with the experimental data at the small tilt angles from the $xy$-plane. However, the disagreement increases with $B_z$, likely due to our calculation approximating side boundary scattering with a constant relaxation time. This is likely a crude approximation because the sample length $L$ is shorter than $W$ in the experiment. The extension of the theory to a realistic sample geometry is straightforward—especially since one may still compute $g$ for every trajectory independently—but it strongly increases the computational costs, and therefore we consider it unjustified for our study.

## 4  Summary

In summary, we demonstrated that the observed magnetoresistance of delafossite materials is explained by the Bloch-like oscillations of the out-of-plane electron trajectories. These Bloch-Lorentz oscillations arise from the quasi-2D dispersion of these materials combined with the nearly ballistic motion of the electrons. We identify the sample aspect ratio as the most likely factor limiting the oscillation visibility. modeling achieves a qualitative agreement with the experiment without introducing any free parameters.

# Acknowledgements

We would like to thank Ady Stern, Veronika Sunko, and Maja Bachmann for the helpful discussions in the early stages of the project. Special thanks to Carlo Beenakker for his valuable feedback and advice on the manuscript. Also, we are grateful to Philip J.W. Moll and Carsten Putzke for sharing with us their experimental findings and subsequent consultations.

**Data Availability**

All code and data used in the manuscript are available at Ref. [21].

**Author contributions**

A.A. formulated the project idea. K.V. developed the theory, carried out numerical simulations, and analyzed the data with input from the other authors. The manuscript was written jointly by K.V. and A.A. with input from L.W.

**Funding information**

This work was supported by the NWO VIDI Grant (016.Vidi.189.180), an ERC Starting Grant 638760, and European Union's Horizon 2020 research and innovation programme FE-TOpen Grant No. 828948 (AndQC).

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
