# Peer review of "Bloch-Lorentz magnetoresistance oscillations in delafossites"

_SciPost Physics, doi:SciPost Phys. 15, 019 (2023)_

## Round 2 · Referee Report · Anonymous (Referee 1) · 2021-8-29

Report

Vilkelis et al. develop a semi-classical theory for the the B-linear oscillations of magnetoresistance observed in the delafossites. This is primarily motivated by the very interesting recent work on PdCoO2 by Putzke et al., Science 368 1234-1238 (Ref. 12 in the manuscript). The basic mechanism postulated by Vilkelis et al. is explained in Fig. 2 a-c: Due to the low disorder in delafossites, electron trajectories out of the plane x-y plane can be commensurate with the sample width (A in Fig. 2) such that a minimum in z-conductivity occurs or in the opposite case of half-integer period (B in Fig. 2) the maximum of conductivity occurs. The authors then also discuss various scattering mechanisms that they claim can more accurately reproduce the experimental data. They also discuss the impact of tilting the magnetic field.

I find the manuscript is mostly well written (with a few minor typos e.g. lenght) and the method of calculation is clear. The calculation is relatively straightforward and appears technically correct for the assumptions they make but I have several concerns whether it actually applies to the system in question (or any other system), I outline these below. I must say from the start that — whilst I do believe that the work is potentially publishable in some form — unfortunately I do not think that it meets the high quality, originality, or interest standards that I would expect of SciPost Physics. I find the ultimate message of the current manuscript rather confused. I am therefore very cautious to recommend publication in any SciPost journal until the authors respond to my queries and much more clearly explain the points of agreement and disagreement with theory and experiments of Putzke et al. in order for the reader to understand what is gained by this study.

Outline of my main questions/concerns:

1) I find the motivation for the manuscript rather confusing. The authors claim that Putzke et al.’s theory “reproduces the results but offers limited insight into the nature of the phenomenon”. To be quite frank I think this statement misleading. Putzke et al. explain clearly the underlying physics of their theory in terms of a diffraction-like grating effect of the hopping between adjacent layers, based on this picture they then perform theoretical calculations using the Kubo formula and it fits very well with their experimental findings (Fig. 4 of Ref. 12). Putzke et al. do admit they are open to alternative explanations at the end of their manuscript, however, the current authors should clearly explain: 
 i) What additional mechanistic insights are gained from their semi-classical explanation? 
ii) What experimental observations can their calculation explain which the mechanism of Putzke et al. cannot? 
 iii) Conversely, are there aspects of the experiment that Putzke et al.’s theory can better explain? 
iv) Do the two theoretical pictures connect with each other or are they contradictory?

2) The authors admit that a ballistic mean free path e.g. the ~ 20 um found in Ref. [8] would rule out their mechanism since the oscillations suggested by the authors would become “fully visible” in such a case. The authors then propose a workaround which is scattering from the sample walls. I think the authors should explain the basis on which they claim this is the “dominant mechanism” for scattering. To be precise, do they believe there is any experimental data in Putzke et al. to support this assertion? Further, how do the authors reconcile their claim with the experimental observation (pg. 4 end - start pg. 5 of Ref. 12) that in samples where the mfp was reduced from 20 um to 1 um by irradiation oscillations were not visible in 8 um wide samples but became visible only when the sample was narrowed to 1 um (i.e. ballistic)? (If boundary scattering is dominant then why does a narrower sample exhibit the oscillatory behaviour but a wider one does not?)

3) The authors have a discussion on titled magnetic fields which concludes that the fact that oscillations disappear when the cyclotron orbits fit into the sample. I was surprised to see that at no point in section 3.3 do the authors mention that Putzke et al. also discuss this and indeed show very nice data of the transition from linear in B oscillations to SdH like oscillations when the field angle is tilted. The authors should outline why their discussion on this matter differs from Putzke et al. or, if the mechanism is the same (which is appears to me they are), they should acknowledge this fact.

4) At no point is the reader shown what “full visibility” of the purported oscillations in conductance look like for a clean system, how the oscillations looks with decreasing mfp,

 or the difference in appearance of the oscillations with/without the dominant boundary scattering. All of these are key messages of the paper.

5) The authors state: "The extension of the theory to a realistic sample geometry is straightforward... but it strongly increases the computational costs and therefore we consider it unjustified for the purposes of our study." I do not understand why the numerical integration of Eq. (29) becomes so costly for a realistic sample geometry and, since the whole message of the manuscript is that the the authors can example the experimental data, it seems strange to claim it is unjustified to show their calculation for the correct experimental geometry.

6) At the end of the abstract the authors state: "Our theory offers a way to design an experimental geometry that is better suited for probing the phenomenon and to investigate the out-of-plane dynamics of ballistic quasitwo-dimensional materials." However, I do not see any discussion of this in the main text. Do the authors simply mean that one should change the aspect ratio L/W to see clearer oscillations, as in Fig. 3?

To conclude: The authors make strong claims about their mechanisms relation to Putzke et al.'s experiments but it is unclear to me what is gained from their study compared to the previous proposed mechanism or how these relate. I do not see the significant advance in understanding of the Putzke et al. experiment which would warrant publication in SciPost Physics. Indeed it is not entirely clear to me whether this (relatively) simple calculation is even able to explain the experiment equally as well as the proposed mechanism by Putzke et al. without resorting to a scattering mechanism which seems unsupported by the actual experiment.

---

## Round 3 · Referee Report · Anonymous (Referee 1) · 2022-10-25

Report

As previously stated: the calculation, although relatively simple, appears technically correct. The authors have now largely addressed my concerns/comments from the previous round and can now be published.

I think the question of the relevance of this semiclassical mechanism in the delafossites considered in Putzke et al. is still open, but would likely require further experiments to properly distinguish between the Kubo mechanism and semiclassical description set out in this paper.

As the authors pointed out, the mechanism is also somewhat similar to a paper by Pippard from 1966 (Ref. 13). Although I think the manuscript will be of some interest to those working on the delafossites, I do not see significant interest for a general audience that would normally be expected of a manuscript in the flagship SciPost Physics journal. I therefore think the manuscript would be better suited to SciPost Physics Core.

---

## Round 3 · Referee Report · Anonymous (Referee 2) · 2022-11-4

Report

This manuscript provides a semiclassical analysis of the delafossite magnetoconductance experiment carried out by Putzke et al.. This had found an oscillatory signal interpreted as an interference effect between electron paths, with the periodicity given by a flux quantum threading a loop of size (width of the sample)*(lattice constant in z-direction).

This appears to me to be a worthwhile study, filling in a gap between the Kubo treatment and the heuristic description provided in the original experimental publication.

One particularly interesting argument relates to the role of surface scattering, and a prediction of an enhanced signal for a differently chosen in-plane aspect ratio.

I have little of substance to add to the report of the other referee. At the same time, it strikes me that the present manuscript overstates the contrast to the publication by Putzke, and that there may be an issue of wording which differs between communities.

Concretely, the authors state that "our formalism does not rely on phase coherence". I suspect that many physicists would consider any interference phenomenon to be based on phase coherence on some level.

I had a similar feeling of unease concerning the statement that "Our result also bears physical interest because it is a rare example of a case where ℏ appears without interfering paths." On some level, I can think of many instances where hbar appears without the need to invoke interfering paths, at least (and perhaps at most) superficially.

In a similar vein, I am not sure I can make any real sense of the authors' statement that "The possible semiclassical origin of the oscillations was overlooked in the previous analysis. "

It seems to me that much of the distinction built up between Putzke et al and the present manuscript is semantic rather than substantial. I would leave it up to the authors how to deal with this. 1- But my recommendation would be to desist from building up expectations about deep insights which dissolve upon closer inspection.

Finally, I have not come across the term 'Lorentz-Bloch oscillations' in the title. The closest to a definition of this term I have found is after Eq. 24. I apologise if I have missed it.
2- but a crisp definition somewhere may not go amiss.

Overall, I recommend an appropriately amended version in scipost core.

Requested changes

See 1- and 2- above.

---

## Round 3 · Author Response

We thank the referee for their report. The referee identified several problems with the presentation of the manuscript, most importantly its relation to the Kubo theory presented in the work of Putzke et al., as well as the benefits it adds. We have critically reassessed the manuscript and we agree with the reviewer that it could be significantly improved, which we did our best to do now. The most important changes in the manuscript are given in the list of changes.

Below we also provide the answers to the specific queries by the referee.

1) I find the motivation for the manuscript rather confusing. The authors claim that Putzke et al.’s theory “reproduces the results but offers limited insight into the nature of the phenomenon”. To be quite frank I think this statement is misleading. Putzke et al. explain clearly the underlying physics of their theory in terms of a diffraction-like grating effect of the hopping between adjacent layers, based on this picture they then perform theoretical calculations using the Kubo formula and it fits very well with their experimental findings (Fig. 4 of Ref. 12). Putzke et al. do admit they are open to alternative explanations at the end of their manuscript, however, the current authors should clearly explain: i) What additional mechanistic insights are gained from their semi-classical explanation? ii) What experimental observations can their calculation explain which the mechanism of Putzke et al. cannot? iii) Conversely, are there aspects of the experiment that Putzke et al.’s theory can better explain? iv) Do the two theoretical pictures connect with each other or are they contradictory?"

By definition, a Kubo approach constructed with the full Hamiltonian is exact and sufficient to explain the complete behaviour of this system. We, therefore, do not dispute the validity of the Kubo approach. On the other hand, the full Kubo approach would require simulating the full sample cross-section down to the wavelength resolution, which is far beyond modern computation capacity.

Additionally, the Kubo approach requires a derivation of the transition rates, which was not performed in the work of Putzke et al. Our approach identifies the Fermi wavelength as a parameter that is irrelevant to the phenomenon and demonstrates that this physics occurs in a purely classical system. We, therefore, identify an important simplification of the Kubo model. Naturally, if for example, quantum oscillations were the goal, our approach would not capture those, while the Kubo formalism would.

Because our theory is semiclassical, it proves that the oscillations do not rely on phase coherence beyond the lattice scales. This is compatible with the persistence of oscillations in an unusually high temperature observed by Putzke et al. The possible semiclassical origin of the oscillations was overlooked in the previous analysis. Specifically the abstract states: "These results demonstrate extraordinary single-particle quantum coherence lengths in the delafossites". We have also found out that similar magnetoresistance oscillations were observed in millimeter-sized samples of ballistic gallium at 1.3K in Ref 14. This serves as additional evidence against the coherent nature of the phenomenon.

Our result also bears physical interest because it is a rare example of a case where $\hbar$ appears without interfering paths. This happens because the quantum mechanical nature of the system is encoded in its periodic Fermi surface. Our approach also allows us to directly model both the sample geometry and the relative impact of different scattering mechanisms.

2) The authors admit that a ballistic mean free path e.g. the ~ 20 um found in Ref. [8] would rule out their mechanism since the oscillations suggested by the authors would become “fully visible” in such a case. The authors then propose a workaround which is scattering from the sample walls. I think the authors should explain the basis on which they claim this is the “dominant mechanism” for scattering. To be precise, do they believe there is any experimental data in Putzke et al. to support this assertion? Further, how do the authors reconcile their claim with the experimental observation (pg. 4 end - start pg. 5 of Ref. 12) that in samples where the mfp was reduced from 20 um to 1 um by irradiation oscillations were not visible in 8 um wide samples but became visible only when the sample was narrowed to 1 um (i.e. ballistic)? (If boundary scattering is dominant then why does a narrower sample exhibit the oscillatory behaviour but a wider one does not?)

Upon rewriting the manuscript we realized that this part of the analysis was not explained clearly. We summarize the argument below. The experiments of Putzke et al. were performed in an effectively ballistic sample with a small aspect ratio. Our model applies to this case and establishes that the aspect ratio explains the observed magnetoresistance. If the sample had a mean free path shorter than its width, as described in Ref. 12, then the bulk scattering would become the dominant mechanism determining the oscillation visibility. Therefore the situation described in the referee's query is in complete agreement with our analysis. We do not claim the contrary in the current version, nor did we in the previous submission.

3) The authors have a discussion on titled magnetic fields which concludes that the fact that oscillations disappear when the cyclotron orbits fit into the sample. I was surprised to see that at no point in section 3.3 do the authors mention that Putzke et al. also discuss this and indeed show very nice data of the transition from linear in B oscillations to SdH like oscillations when the field angle is tilted. The authors should outline why their discussion on this matter differs from Putzke et al. or, if the mechanism is the same (which is appears to me they are), they should acknowledge this fact

Section 3 of our manuscript compares the predictions of the semiclassical theory with the measurement reported in the work of Putzke et al. The authors have indeed identified the relevance of cyclotron radius in controlling oscillation appearance, which we now clearly acknowledge in the manuscript, and we apologise for not clearly stating this earlier. The relation between the two approaches is the same in this case as we have explained previously.

4) At no point is the reader shown what “full visibility” of the purported oscillations in conductance look like for a clean system, how the oscillations looks with decreasing mfp, or the difference in appearance of the oscillations with/without the dominant boundary scattering. All of these are key messages of the paper.

The full visibility oscillations are given by the Eq. 22 of the manuscript, which has $\sigma_{zz} \propto (1 - \cos(\omega B_y))$. We believe that this function is sufficiently simple to not require a plot. The previous version of the manuscript demonstrated the effect of the sample aspect ratio in Fig. 3, which we have extended with a similar plot for the mean free path in the resubmitted version.

5) The authors state: "The extension of the theory to a realistic sample geometry is straightforward... but it strongly increases the computational costs and therefore we consider it unjustified for the purposes of our study." I do not understand why the numerical integration of Eq. (29) becomes so costly for a realistic sample geometry and, since the whole message of the manuscript is that the the authors can example the experimental data, it seems strange to claim it is unjustified to show their calculation for the correct experimental geometry.

We have now explained that combining the full sample geometry with an out-of-plane magnetic field requires a 4D integral. While it is possible to carry it out, unlike a fully quantum-mechanical simulation on the scale of the Fermi wavelength, we believe that the added benefit is minimal, and therefore model this approximately. This approximation is qualitatively appropriate, as demonstrated by Figure 3.

6) At the end of the abstract the authors state: "Our theory offers a way to design an experimental geometry that is better suited for probing the phenomenon and to investigate the out-of-plane dynamics of ballistic quasitwo-dimensional materials." However, I do not see any discussion of this in the main text. Do the authors simply mean that one should change the aspect ratio L/W to see clearer oscillations, as in Fig. 3?

We indeed referred to the aspect ratio, however upon revisiting the manuscript we have identified this statement as hard to interpret, and therefore removed it.

---

## Round 3 · List of Changes

• We clarified the relationship to the Kubo theory, in particular, explaining that Kubo theory is correct and that our approach extends that understanding by identifying the phenomenon as a consequence of the open trajectories and identifying the sample boundary scattering as the main mechanism limiting the intensity of the magnetoresistance oscillations.
  • We have reorganized the derivation to make it more universal and straightforward to follow. In particular, we demonstrated that in a general setting the magnetoresistance response is described by the independent behavior of in-plane trajectories.
  • We have restructured the discussion, and clearly state that it is the sample geometry that likely determines the magnetoresistance oscillations amplitude.
  • We identified and included earlier works observing related phenomena in crystalline gallium in Ref 14 as well as a qualitative explanation in Ref 13.

---

## Round 4 · Referee Report · Anonymous (Referee 2) · 2023-4-5

Report

Of the three statements I criticised, one has been withdrawn, the other corrected, and the third is now emphasized with increased vigour.

The latter point strikes me as reasonable, in that the authors take up a form of words utilised in the publication which motivated their study. I had not properly appreciated this aspect in my preceding report.

In my preceding report, I had made the following suggestion: "It seems to me that much of the distinction built up between Putzke et al and the present manuscript is semantic rather than substantial. I would leave it up to the authors how to deal with this. 1- But my recommendation would be to desist from building up expectations about deep insights which dissolve upon closer inspection."

I think how the authors have dealt with the first statement has addressed that point. Regarding the second, their response states:

"our model requires being able to resolve the bandwidth of the interlayer hopping tz≈10 meV∼100 K, as opposed to assuming phase coherence over the scale of 10 microns at similar temperatures. "

I am unable to process the logical status of the 'as opposed' in the middle of the sentence, in as far as I do not see an opposition between the two statements. This reinforces my impression that the thinking of the authors contains further assumptions and/or semantics which are not shared by me, and perhaps other members of the community.

But I think this refereeing process has now run its course, and I am happy to let the readers make up their own mind, as the strong points of the manuscript in my eyes are clearly in favor of publication.

---

## Round 4 · Referee Report · Anonymous (Referee 1) · 2023-4-6

Report

In their response arguing for publication in Scipost Physics the authors’ put a heavy emphasis on the fact Putzke et al. was published in Science. Stating their “work invalidates a critical logical step supporting this claim, which is the main reason why we believe it fulfills the requirements of SciPost Physics.” If the authors truly believe that the claims of the Science paper are absolutely not correct, then the appropriate mechanism would be to write a comment on that paper.

In my mind the authors’ simply provide a (technically correct) semiclassical calculation, very similar to Pippard 1966, which may or may not have anything to do with the Putzke experiment. If the authors wanted to make such a strong contrast with the experiment, a constructive course of debate would have been to state the advantages/disadvantages of each explanation and then to suggest experiments to distinguish between the mechanisms. In contrast, the authors now stating that the existence of their explanation “invalidates” the original explanation seems premature without further experimental evidence. (I note that the authors have a new preprint in this direction, but I can only judge the paper that I was first sent to review almost 2 years ago).

The one piece of evidence the authors claim supports their explanation is that the coherence length measured using SdH oscillations is an order of magnitude shorter than observed in B-periodic oscillations. However, Putzke et al. already explain that the relevant coherence length for SdH oscillations is averaged over the full hexagonal FS whereas the B-periodic oscillations only arise from the flat portions of the FS. This seems a quite reasonable explanation to me that is backed up by the results of Nandi et al. 2018.

To conclude: I do not think that this paper “invalidates” the previous Science paper. It does provide an explanation that requires further experiments to distinguish between the original explanation and this one. Therefore, as previously stated, it is certainly of interest to the Delafossite community, but until such experiments are performed it is of limited interest to those outside of the community (especially as the mechanism was already discussed by Pippard). I stick with my original suggestion that this paper should be published in SciPost Physics Core.

A final thing, I note in the latest version the final sentence has a typo “. modelingling”.

---

## Round 4 · Author Response

We thank the referees for their feedback. The summary of the revisions is in the list of changes below.

We would also like to directly respond to some of the referees' objections. While the referees raise no concerns regarding the technical validity of our work, they question the impact of our work and its suitability to SciPost physics. Before responding to the detailed points raised by the referees, we would like to address this general evaluation.

The manuscript by Putzke et al. reports an "extraordinary" long range and high temperature quantum phase coherence in delafossites. This claim is central to the manuscript: it is the focus of the popular summary, abstract, and the conclusion. For example:

  • The abstract states "These results demonstrate extraordinary single-particle quantum coherence lengths in delafossites."
  • In explaining the new finding of the manuscript, the second paragraph states "Here, we report an unexpected robust manifestation of phase coherence intrinsic to the out-of-plane transport" and lists no other new observations.
  • The final sentence of the paper is "As quantum coherence emerges as its own subject in technology, it will be interesting to explore whether applications can exploit the rare macroscopic, single-particle phase coherence in the delafossites."

That this claim is indeed viewed as extraordinary by the community is confirmed by the manuscript being published in Science (the other properties of delafossites reported in the manuscript were known before). The reasoning of the manuscript critically relies on semiclassical explanations of the phenomenon not existing. In fact, the authors examine several possible explanations and rule them out, concluding that no semiclassical explanations exist.

Because our work proposes such a semiclassical explanation, it removes the need to explain the unexpected behavior. Our model requires no additional assumptions beyond established facts about delafossites, namely their high ballistic mean free path. This is the main impact of our work, and the reason why we consider it relevant to the community.

In the resubmitted version we have clarified this relation between our result and the Ref. 12.

Response to specific referee remarks

I have little of substance to add to the report of the other referee. At the same time, it strikes me that the present manuscript overstates the contrast to the publication by Putzke, and that there may be an issue of wording which differs between communities.

Concretely, the authors state that "our formalism does not rely on phase coherence". I suspect that many physicists would consider any interference phenomenon to be based on phase coherence on some level.

Our statement was indeed not technically accurate: we do assume that the band structure of the material exists, and that is a phase coherent phenomenon. We, however, disagree with the referee's assessment that we overstate the contrast: our model requires being able to resolve the bandwidth of the interlayer hopping $t_z \approx 10 \textrm{ meV} \sim 100 \textrm{ K}$, as opposed to assuming phase coherence over the scale of 10 microns at similar temperatures. In the updated version of the manuscript we stated specifically that our explanation does not rely on phase coherence of trajectories traversing the sample.

I had a similar feeling of unease concerning the statement that “Our result also bears physical interest because it is a rare example of a case where ℏ appears without interfering paths.” On some level, I can think of many instances where hbar appears without the need to invoke interfering paths, at least (and perhaps at most) superficially.

While we do not see a problem with the statement—in most cases $\hbar$ does appear in presence of interfering paths, we have removed the statement from the updated manuscript because it is not central to our claim.

In a similar vein, I am not sure I can make any real sense of the authors' statement that "The possible semiclassical origin of the oscillations was overlooked in the previous analysis. "

In the work by Putzke et al, the central long-range phase coherence claim relies on the absence of semiclassical theories which would explain the observed phenomenon. Below we quote verbatim the paragraph from the manuscript, where the authors rule out possible semiclassical origins of the observed phenomenon.

The few known B-linear oscillatory phenomena in singly connected solids are semiclassical [Sondheimer resonances (16), Azbel-Kaner cyclotron motion (17), or geometric resonances in the presence of acoustic waves]; rely on a superconducting order parameter to establish macroscopic phase coherence [Fraunhoffer interference in Josephson junctions (18)] or exploit artificially introduced nanometric length scales or interference effects in tunneling between parallel quantum wires or wells (19). None of these can explain our data because the observed h/e periodicity clearly indicates long-range, single-particle-phase coherence as their origin.

This quote fully substantiates our claim.

It seems to me that much of the distinction built up between Putzke et al and the present manuscript is semantic rather than substantial.

The claim of "extraordinary" long range phase coherence due to the observations that cannot be explained by a semiclassical mechanism and a semiclassical explanation of the same observation are substantially different.

I think the question of the relevance of this semiclassical mechanism in the delafossites considered in Putzke et al. is still open, but would likely require further experiments to properly distinguish between the Kubo mechanism and semiclassical description set out in this paper.

We agree that a further experiments would be required to determine whether delafossites have such long range phase coherence. For example, we argue in our latest work (arXiv:2303.04310) that if delafossites have long-range coherence then diamagnetic oscillations with the same Aharonov-Bohm period should appear.

As the authors pointed out, the mechanism is also somewhat similar to a paper by Pippard from 1966 (Ref. 13). Although I think the manuscript will be of some interest to those working on the delafossites, I do not see significant interest for a general audience that would normally be expected of a manuscript in the flagship SciPost Physics journal. I therefore think the manuscript would be better suited to SciPost Physics Core.

The claim of unusual phase coherence is interesting to a broad physics community. The work by Putzke et al. was clearly evaluated to be of broad interest because it is published in Science. Our work invalidates a critical logical step supporting this claim, which is the main reason why we believe it fulfills the requirements of SciPost Physics.

---

## Round 4 · List of Changes

* In the introduction, we add Ref[13] to show how unusual high-temperature long-range coherence claimed by Ref[12] is.
* Section 2 added the temperature upper bound for which the derived semiclassical result is valid.
* "Bloch-Lorentz oscillations" term defined at the end of section 3.1.

---

## Editorial Decision

published